# Prediction Model including Gastrocnemius Thickness for the Skeletal Muscle Mass Index in Japanese Older Adults

**DOI:** 10.3390/ijerph19074042

**Published:** 2022-03-29

**Authors:** Satoshi Yuguchi, Ryoma Asahi, Tomohiko Kamo, Masato Azami, Hirofumi Ogihara

**Affiliations:** 1Department of Physical Therapy, School of Health Sciences, Japan University of Health Sciences, Saitama 340-0145, Japan; r-asahi@jhsu.ac.jp (R.A.); t-kamo@jhsu.ac.jp (T.K.); m-azami@jhsu.ac.jp (M.A.); 2Division of Physical Therapy, Department of Rehabilitation, Faculty of Health Sciences, Nagano University of Health and Medicine, Nagano 381-2227, Japan; ogihara.hirofumi@shitoku.ac.jp

**Keywords:** older adults, skeletal muscle mass index, ultrasonography, gastrocnemius thickness, prediction model

## Abstract

Non-invasive and easy alternative methods to indicate skeletal muscle mass index (SMI) have not been established when dual energy X-ray absorptiometry (DXA) or bioelectrical impedance analysis (BIA) cannot be performed. This study aims to construct a prediction model including gastrocnemius thickness using ultrasonography for skeletal muscle mass index (SMI). Total of 193 Japanese aged ≥65 years participated. SMI was measured by BIA, and subcutaneous fat thickness and gastrocnemius thickness in the medial gastrocnemius were measured by using ultrasonography, and age, gender and body mass index (BMI), grip strength, and gait speed were collected. The stepwise multiple regression analysis was conducted, which incorporated SMI as a dependent variable and age, gender, BMI, gastrocnemius thickness, and other factors as independent variables. Gender, BMI, and gastrocnemius thickness were included as significant factors, and the formula: SMI = 1.27 × gender (men: 1, women: 0) + 0.18 × BMI + 0.09 × gastrocnemius thickness (mm) + 1.3 was shown as the prediction model for SMI (R = 0.89, R^2^ = 0.8, adjusted R^2^ = 0.8, *p* < 0.001). The prediction model for SMI had high accuracy and could be a non-invasive and easy alternative method to predict SMI in Japanese older adults.

## 1. Introduction

Sarcopenia, characterized by low skeletal muscle mass, leads to the decline of physical performances and malnutrition, and is associated with adverse events, hospitalization, and reduced life expectancy [1,2,3]. The 2019 Asian Working Group for Sarcopenia (AWGS) recommended diagnosing sarcopenia based on evaluating the skeletal muscle mass in a whole body by dual-energy X-ray absorptiometry (DXA) or bioelectrical impedance analysis (BIA) and physical function, such as grip strength, 5-chair stand test, and walking speed [4] to prevent progressing sarcopenia. Sarcopenia assessments can be conducted in any situation because they are non-invasive and relatively easy. However, in several cases, skeletal muscle mass measurement by DXA and BIA cannot be performed. A previous study reported that DXA and BIA accuracy declines in patients with generalized edema or metal or any devices implanted. BIA is contraindicated in patients with a cardiac pacemaker; moreover, it is difficult to transfer a laboratory in the case of patients with limited physical activity owing to unstable health conditions [5]. Furthermore, the abovementioned patients frequently have chronic diseases, such as heart failure and pulmonary diseases, and have high risks for a decline in skeletal muscle mass [6]. The decline in skeletal muscle mass affects life expectancy in patients with chronic diseases [4,7,8], and assessments and interventions to prevent the occurrence of low skeletal muscle mass are required. However, an alternative assessment has not been established when DXA or BIA cannot be performed.

Recently, several studies, which measured the skeletal muscle thickness by using ultrasonography and examined the association of the thickness with the skeletal muscle mass in a whole body, have been practiced, and the consensus of the European Working Group on Sarcopenia in Older People on the definition and diagnosis refers to a possibility of ultrasonography as an evaluation tool for skeletal muscle mass [8]. The previous study reported that the quadriceps thickness measured by ultrasonography showed a significant difference between patients with and without sarcopenia [9]. Further, the gastrocnemius thickness was related to the low skeletal muscle mass [10,11], and the calf muscle thickness was reportedly the most associated with the skeletal muscle mass in extremities [12]. Conversely, the previous study showed that skeletal muscle mass could be affected by age, gender, body mass index (BMI), and physical performances [13]. However, an alternative method to predict skeletal muscle mass using muscle thickness measured by ultrasonography is unknown, considering the factors related to skeletal muscle mass. We hypothesized that muscle thickness measured by ultrasonography could predict skeletal muscle mass and become an alternative, non-invasive, and relatively easy method when DXA or BIA could not be performed.

This study aims to measure gastrocnemius thickness using ultrasonography and construct a prediction model, including gastrocnemius thickness and other factors, for skeletal muscle mass.

## 2. Materials and Methods

### 2.1. Study Design and Participants

This cross-sectional study measured whole body skeletal muscle mass by using ultrasonography among community-dwelling healthy Japanese older adults aged ≥65 years living independently in Satte City, Saitama prefecture, Japan. This study was conducted from May to December 2019. A total of 210 participants were recruited in this study, with the following exclusion criteria: those with sarcopenia, and severe diseases, such as stroke, neuromuscular disease, and cardiac disease; those with a walking aid; and those with unmeasured data. Finally, this study included 193 Japanese older adults. It complied with the guidelines proposed by the Declaration of Helsinki. The aim and protocol of this study were reviewed and approved by the ethics committee of Japan University of Health Sciences (approval number: 3001).

### 2.2. Data Collection

The skeletal muscle mass was measured using the bioelectrical impedance system (MC-780; TANITA, Corp, Tokyo, Japan). All participants were instructed not to eat breakfast, and measurements were conducted from 8:00 to 11:00 AM at the same place. This system uses an electric current at different frequencies (5, 50, and 250 kHz) to measure the amount of extra- and intracellular fluid in the body. During the test, the participants were asked to stand on two metallic electrodes and hold metallic grip electrodes, and a value for the appendicular skeletal muscle mass (kg) was measured. The appendicular skeletal muscle mass was converted into skeletal muscle mass index (SMI) by dividing the muscle weight by the height squared (kg/m^2^) [9]. SMI was analyzed in this study. Low SMI was defined as <7.0 and <5.7 kg/m^2^ in men and women, respectively [4].

An ultrasound device (View’s i; MINATO Medical Science Co., Ltd., Osaka, Japan) with a 6-MHz linear array probe was used to obtain an image of the gastrocnemius muscle. The ultrasonography setting was consistently set at B-mode and default and fixed dB dynamic range and gain. The participants were assessed while sitting and with the knee joint flexed at 90° and the ankle joint at 0°. While checking the A-mode of the device monitor, the same examiner vertically and lightly placed the probe on the right medial gastrocnemius in the maximum part of the below-knee circumference. The intra-rater reliability was reportedly lower when the pressure was <100 gf [14]. The probe pressure on the skin was controlled at 200 gf. The examiner scanned the image of the subcutaneous adipose tissue and gastrocnemius muscle (Figure 1). Subcutaneous fat thickness (mm) was defined as the distance between the surface and upper fascia of the gastrocnemius; gastrocnemius thickness (mm) was defined as the distance between the subcutaneous to the deep fascia. Subsequently, subcutaneous fat and gastrocnemius thicknesses were measured.

The reproducibility of the gastrocnemius thickness was measured in the first 17 participants using the intra-class correlation coefficient (ICC). ICC (1.2) and (1.1) of the gastrocnemius thickness was >0.96; therefore, the number of measured ultrasound images was defined once.

Grip strength and walking speed were measured following the AWGS diagnostic criteria [4], and grip strength and walking speed were measured once after pretest. Grip strength was measured using a digital dynamometer (T.K.K.5401; Takei Corp., Japan) in an upright position and defined as the highest value in the right or left hand for each trial. The walking speed (m/s) was measured and defined as speed in walking the 6-m distance at a comfortable pace. Low grip strength was defined as <28 and <18 kg in men and women, respectively, and low gait speed was defined as <1.0 m/s in both men and women [4].

### 2.3. Statistical Analysis

In this study, normality was assessed using the Shapiro–Wilk test. Pearson correlation coefficient was analyzed to determine the relationship between SMI and age, BMI, grip strength, gait speed, and subcutaneous fat and gastrocnemius thicknesses, respectively. A stepwise multiple regression analysis which is a method of constructing regression models by automatically choosing only predictive variables from independent variables was performed to construct the prediction model for SMI. The analysis incorporated SMI as the dependent variable and age, gender, BMI, grip strength, gait speed, subcutaneous fat and gastrocnemius thicknesses as independent variables. Moreover, the residual analysis of the prediction model for SMI was analyzed using the Durbin–Watson test. All data were analyzed using IBM SPSS Statistics Version 27. The significance level was set to *p* < 0.05 for all tests.

## 3. Results

Figure 1 shows the image of subcutaneous fat and gastrocnemius thickness by ultrasonography.

Table 1 shows the participant characteristics in terms of age, BMI, physical performances, SMI, and ultrasonography (men: *n* = 72; women: *n* = 121). In this study, the mean age was 72.4 years, and in comparison, men showed significantly higher age, BMI, grip strength, SMI, and gastrocnemius thickness than women, and the subcutaneous thickness of women was significantly higher than that of men.

Table 2 shows the univariate analysis results for the correlation of SMI with age, gender, BMI, grip strength, gait speed, and subcutaneous and gastrocnemius thicknesses, respectively. SMI showed significant correlations with BMI (r = 0.67), grip strength (r = 0.62), and gastrocnemius thickness (r = 0.51), respectively (*p* < 0.001).

Table 3 shows the result of the stepwise multiple regression analysis that incorporated SMI as the dependent variable. In the analysis, men, BMI, and gastrocnemius thickness were included as significant factors in models 1, 2, and 3, respectively. Conversely, age, grip strength, gait speed, and subcutaneous thickness were excluded as significant factors. Finally, the stepwise multiple regression analysis showed the prediction model for SMI as the below formula:SMI = 1.27 × gender (men: 1, women: 0) + 0.18 × BMI + 0.09 × gastrocnemius thickness (mm) + 1.3.

The value of adjusted R^2^ was 0.8, and the prediction model’s contribution rate, including gastrocnemius thickness for SMI was high. The statistic value of the Durbin–Watson test was 1.65 which was near 2, and only one participant was not included within ± 3SD (Figure 2). In addition, Figure 3 shows the plots of the actual SMI and unstandardized predicted SMI, and the validity of the prediction model was high.

## 4. Discussion

This is the first study that constructed the prediction model for SMI, including the gastrocnemius thickness by ultrasonography as an independent variable. The accuracy of the prediction model for SMI was high.

Regarding the participant characteristics, age and BMI showed significant differences between men and women; however, there were no differences between the proportions of men and women with low grip strength, low gait speed, and low SMI. Compared with the previous study on Japanese older adults [8], the participants had better physical performances. In the ultrasonography findings, the subcutaneous thickness of men was lower than that of women, and the gastrocnemius thickness of men was higher than that of women. AWGS 2019 has reported that the normal values of the calf circumference and SMI are >34.0 cm in men and >33.0 cm in women, and >7.0 kg/m^2^ in men and >5.7 kg/m^2^ in women, respectively [4]. Additionally, the previous studies showed that the subcutaneous thickness of men was lower than that of women, and the skeletal muscle thickness of men was higher than that of women as measured by ultrasonography [9,15,16,17]. In this study, the abovementioned result of gender differences in ultrasonography was consistent with the results of the previous studies.

This study showed that gender, BMI, and gastrocnemius thickness were included as factors associated with SMI, and age, grip strength, gait speed, and subcutaneous thickness were excluded. Regarding the factors related to SMI, in 400 Asian people over the age of 65 years, gender, age, BMI, and calf circumference were associated with SMI; however, age was not associated with SMI after adjustment for gender [18]. Furthermore, the previous study reported that in 616 South American people over the age of 60 years, the multiple regression analysis, which incorporated SMI as the dependent variable, and gender, age, weight, knee height, grip strength, and calf and hip circumferences as independent variables, resulted in the contribution rate of 0.89 and a high value [13]. While the results of several previous studies supported gender and BMI as factors of SMI, the result of our study wherein age was excluded as a factor of SMI was consistent with that of a previous study with the same age and Asian people as participants [18]. However, another study of Japanese older adults reported that despite having a normal SMI, participants had lower physical performances such as grip strength and gait speed, compared to the normality [8]. Racial difference, therefore, could affect our results excluding age and physical performance. Moreover, the previous study reported that physical performances were not necessarily related to SMI [9], physical performances could have been excluded as factors of SMI. In this study, gender was included as a factor of SMI, and the result of excluding grip strength as the factor could be affected by gender as a confounding factor because grip strength between gender had difference in Table 1.

Here, the gastrocnemius thickness by ultrasonography was included as a factor of SMI. A previous study in Asian and European populations reported that measuring SMI and fat-free mass index, which reflected the skeletal muscle mass in a whole body, the gastrocnemius thickness by ultrasonography was significantly associated with low skeletal muscle mass [9,10]. Further, calf circumference has been used to screen skeletal muscle mass [4]. Therefore, because it is probable to infer that the gastrocnemius thickness, which affects the calf circumference, is strongly related to the skeletal muscle mass, it could have been selected as an independent variable associated with SMI.

Here, we constructed the prediction model for SMI including three independent variables, such as gender, BMI, and gastrocnemius thickness by ultrasonography. A previous study in South America reported that the contribution rate of the prediction model for SMI calculated by seven independent variables showed a high value of 0.89 [13]. It would be a useful method for predicting SMI because the prediction model was constructed by noninvasively measurable variables, such as calf circumference, knee height, and grip strength. However, owing to the inclusion of grip strength, the participants were required to cooperate with the measurement and have no cognitive function and stable general conditions, and the prediction model included the calf circumference, which was affected by generalized or leg edema. Moreover, the prediction model was limited to the Asian population due to South American participants’ examination; however, a model of SMI for Asian people has not yet been constructed. The prediction model in our study targeted Japanese people over the age of 65 years and was consisted of three non-invasive and relatively easy parameters. Especially, muscle thickness by ultrasonography reflected the change of the skeletal muscle thickness for patients with unstable conditions, fluid imbalances, generalized edemas, and on a mechanical ventilator in intensive care units (ICUs) [19]. Therefore, the prediction model constructed in our study could deal with the past challenges such as difficulty in measuring DXA and BIA in patients with generalized edema, metal devices, or a cardiac pacemaker implanted, and with limited physical activity owing to unstable health conditions [5]. In addition, ultrasonography device has become more affordable and portable recently. The prediction model is expected to predict SMI noninvasively and easily when BIA or DXA could not be performed.

This study has several limitations. First, ultrasonography was measured in the sitting position, which could be different from the measurement posture in other studies. Second, during the measurement of SMI by bioelectrical impedance analysis, factors such as temperature and humidity could not be monitored. Third, as the study included Japanese people over the age of 65, it is unclear whether the prediction model can adapt to people under the age of 64 or non-Asians. Fourth, since the participants had no sarcopenia with relatively good physical performance, the adaptability of the prediction model for participants with sarcopenia is not proven. Finally, it is unclear whether the prediction model can adapt to patients with unstable general conditions, generalized edemas, and on a mechanical ventilator in ICUs.

## 5. Conclusions

To construct the prediction model for SMI, the stepwise multiple regression analysis that incorporated SMI as a dependent variable was performed. Gender, BMI, and gastrocnemius thickness by ultrasonography were selected as significant factors. The prediction model for SMI had high accuracy and could be a non-invasive and relatively easy alternative method to predict SMI in Japanese older adults.

## Figures and Tables

**Figure 1 ijerph-19-04042-f001:**
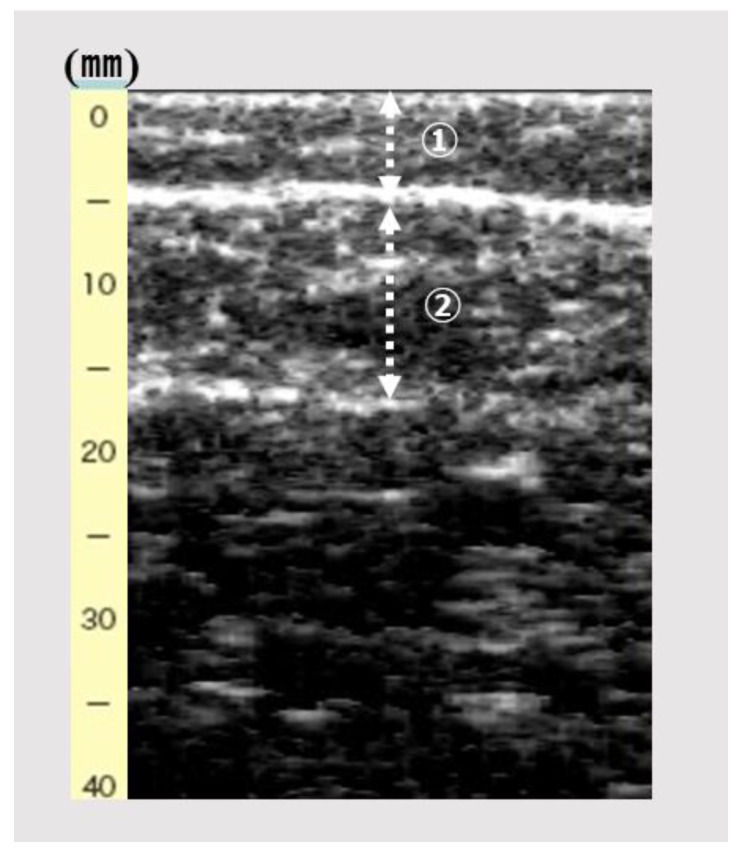
The image of subcutaneous fat and gastrocnemius thickness by ultrasonography. ➀, Subcutaneous fat thickness (mm); ②, gastrocnemius thickness (mm).

**Figure 2 ijerph-19-04042-f002:**
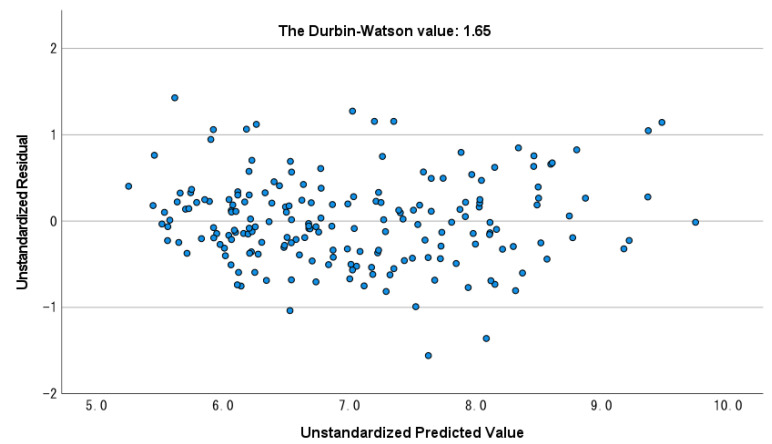
The residual plot and Durbin–Watson value for the dependent variable in stepwise multiple regression analysis.

**Figure 3 ijerph-19-04042-f003:**
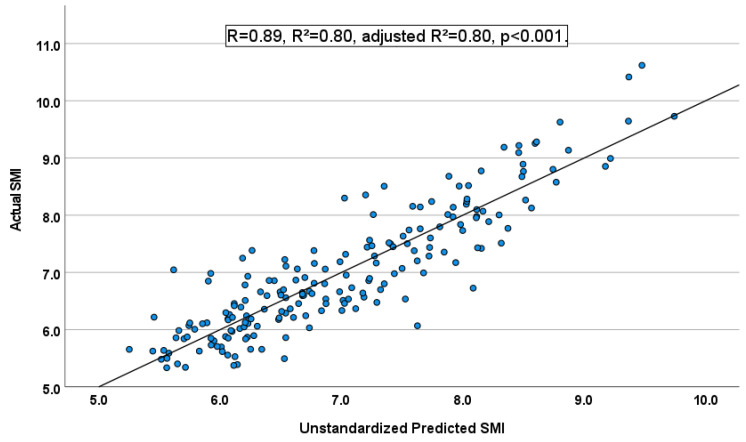
Actual SMI and unstandardized predicted SMI plots calculated by the stepwise multiple regression analysis.

**Table 1 ijerph-19-04042-t001:** Characteristics of participants.

	Total *n* = 193	Men *n* = 72	Women *n* = 121	*p*-Value (Men vs. Women)
Age; years	72.4 ± 4.3	73.2 ± 4.3	71.9 ± 4.2	<0.05
BMI; kg/m^2^	22.4 ± 2.9	23.2 ± 3.0	21.9 ± 2.8	<0.01
Grip strength; kg	28.6 ± 7.9	36.6 ± 5.9	23.8 ± 4.2	<0.001
Low grip strength; n (%)	11 (5.7)	4 (5.5)	7 (5.8)	0.77
Gait speed; m/s	2.0 ± 0.4	1.9 ± 0.4	2.0 ± 0.4	0.24
Low gait speed; n (%)	9 (4.7)	6 (8.3)	3 (2.5)	0.06
SMI; kg/m^2^	7.0 ± 1.1	8.0 ± 1.0	6.4 ± 0.7	<0.001
Low muscle mass; n (%)	31 (16.1)	12 (16.7)	19 (15.7)	0.86
Ultrasonography				
SFT; mm	4.1 ± 2.2	2.6 ± 1.5	5.0 ± 2.1	<0.001
GT; mm	13.0 ± 2.2	13.6 ± 2.6	12.7 ± 1.8	<0.01

BMI, body mass index; SMI, skeletal muscle mass index; SFT, subcutaneous fat thickness; GT, gastrocnemius thickness. Low grip strength was defined as <28 and <18 kg in men and women, and low gait speed was defined as <1.0 m/s in both men and women.

**Table 2 ijerph-19-04042-t002:** Univariate analysis by Pearson correlation coefficient.

Variable	r	*p*
Age	0.15	0.49
BMI	0.67	<0.001
Grip strength	0.62	<0.001
Gait speed	−0.06	0.41
SFT	−0.09	0.22
GT	0.51	<0.001

Dependent variable: SMI (skeletal muscle mass index); BMI, body mass index; SFT, subcutaneous fat thickness; GT, gastrocnemius thickness.

**Table 3 ijerph-19-04042-t003:** Multivariable analysis by stepwise multiple regression analysis.

Variable	Model 1			Model 2			Model 3			
	B	SE B	β	B	SE B	β	B	SE B	β	VIF
Men	1.57	0.12	0.69 *	1.32	0.08	0.58 *	1.27	0.08	0.56 *	1.06
BMI				0.21	0.01	0.55 *	0.18	0.01	0.47 *	1.27
GT							0.09	0.02	0.19 *	1.27
α	6.41	0.07		1.88	0.30		1.33	0.30		
R	0.69			0.88			0.89			
R^2^	0.48			0.77			0.80			
Adjusted R^2^	0.48			0.77			0.80			
F	175.2 *			318.1 *			247.6 *			

Dependent variable: SMI; excluded variables: age, SFT, grip strength, gait speed. B, partial regression coefficient; SE, standard error; β, standardized partial regression coefficient; VIF, variance inflation factor; BMI, body mass index; GT, gastrocnemius thickness. * *p* < 0.001.

## Data Availability

Not applicable.

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
