# Peer review of "Prediction Model including Gastrocnemius Thickness for the Skeletal Muscle Mass Index in Japanese Older Adults"

_ijerph, 2022, doi:10.3390/ijerph19074042_

Round 1
Reviewer 1 Report
In this study Dr. Satoshi et al present a mathematical model for prediction of skeletal mass index, that utilizes gastrocnemius ultrasonography, gender and body mass index. The authors tested three mathematical models and the most efficient reaches an R2 of 0.8 as compared to the bioelectrical impedance method. It is concluded that this model can be used in patients that cannot perform bioelectrical impedance, either because have metal implants or edema. The number of subjects evaluated is appropriated (195) to conduct such study and the correlations found seem solid. Nonetheless, a series of questions need to be addressed as follows
Text needs English language grammar review that in some cases even disturb the proper understanding:
Line 189 - Regarding the participant characteristics, age and BMI showed significant differ-188 ences; however, normal values by gender in physical performances and SMI had no dif-189 ferences
“significant differences” related to what?
“normal” suggest to remove normal and again “had no difference” compared to what
Line 202 - This study showed that gender, BMI, and gastrocnemius thickness were included as 201 factors associated with SMI, and age, grip strength, gait speed, and subcutaneous thick-202 ness were excluded.
Why exclude grip strength since it had a positive correlation (p <0.001) with SMI?
Line 209-212 - While the results of several previous studies supported gender and BMI as factors of SMI, the result of our study wherein age was excluded as a factor of SMI was consistent with that of a previous study with the same age and Asian people as participants.
Please provide references
Line 216-218 - Moreover, because physical performances were not necessarily related to SMI, suggesting that the skeletal muscle mass in a whole body was indicated, physical performances could have been excluded as factors of SMI in our study.
Very confusing phrase, please revise
Methods
Lines 89-90 - The participants were assessed while sitting and with the knee joint flexed at 90° and the ankle joint plantar dorsiflexed at 0°.
Suggest: The participants were assessed while sitting, with the knee joint flexed at 90° and the ankle joint at 0°. Also please add a reference.
Grip strength was measured using a digital dynamometer (T.K.K.5401; Takei Corp., Japan) in an upright position and defined as the highest value in the right or left hand for each trial.
How many was the test performed for each individual? Which one have you considered? Was any adaptation period? Meaning pre-repetitions aiming to get the subject familiar with the movement?
A stepwise multiple regression analysis was performed to construct the prediction model for SMI. The analysis incorporated SMI as the dependent variable and age, gender, BMI, grip strength, gait speed, and subcutaneous fat and gastrocnemius thicknesses as independent variables. Moreover, the residual analysis of the prediction model for SMI was analyzed using the Durbin–Watson test.
Why did you use Drubin-Watson? Please justify
Results
Table 1
Low gait speed and low grip strength is not clear. Please explain and describe it better in material and methods, for example how many tests have you performed?
Table 3
Adjusted R2 is not described neither in the legend nor in material and methods
Why did you not considered grip as a factor in the model? It had a positive relationship with SMI (table 2).
It would be interesting to have the individual values of SMI in parallel with the individual prediction values as a supplementary figure.
Figure 2
Mentioned as just on two lines in the results section and the very same statement is mentioned in the discussion section (was 2.03). What does the Durbin-Watson test mean? What message can the reader take from this test? How does it compare with the other analyses performed?
Author Response
Dear reviewer 1
Thank you very much for a lot of constructive advice for our manuscript in revision process.
I have answered for reviewer comments, added some sentences and revised our manuscript according to reviewer comments, and I have underlined at each sentence which I revised.
1.Line 189 - Regarding the participant characteristics, age and BMI showed significant differences; however, normal values by gender in physical performances and SMI had no differences
“significant differences” related to what?
“normal” suggest to remove normal and again “had no difference” compared to what
→I’m sorry to confuse you. Significant differences mean differences of age and BMI between gender, and “had no differences means the proportions of low physical performances and SMI. So, I revised the sentences in line 189-191 as follows:
Regarding the participant characteristics, age and BMI showed significant differences between men and women; however, there were no differences between the proportions of men and women with low grip strength, low gait speed and low SMI.
2.Line 202 - 201 This study showed that gender, BMI, and gastrocnemius thickness were included as factors associated with SMI, and age, grip strength, gait speed, and subcutaneous thick- ness were excluded.
Why exclude grip strength since it had a positive correlation (p <0.001) with SMI?
→Thank you very much for your comment. The stepwise multiple regression analysis automatically excluded grip strength as a factor of SMI. I think that the result would be affected by racial difference according to the previous studies in line 204-217, and the weak relationship between physical performances and SMI by the previous study in line 217-219. Moreover, gender as a confounding factor would affect the result because grip strength had difference between gender in characteristics of participants. So, I added the sentence into the line 219-222 as follows:
In this study, gender was included as a factor of SMI, and the result of excluding grip strength as the factor could be affected by gender as a confounding factor because grip strength between gender had difference in table 1.
3.Line 209-212 - While the results of several previous studies supported gender and BMI as factors of SMI, the result of our study wherein age was excluded as a factor of SMI was consistent with that of a previous study with the same age and Asian people as participants.
Please provide references
→Thank you very much for your indication. I added reference into line 213 as follows:
While the results of several previous studies … as a factor of SMI was consistent with that of a previous study with the same age and Asian people as participants [18].
4.Line 216-218 - Moreover, because physical performances were not necessarily related to SMI, suggesting that the skeletal muscle mass in a whole body was indicated, physical performances could have been excluded as factors of SMI in our study.
Very confusing phrase, please revise
→ I‘m sorry to confusing you. I revised the sentence and add the reference in line 217-218 as follows:
Moreover, the previous study reported that physical performances were not necessarily related to SMI [9], physical performances could have been excluded as factors of SMI.
- Lines 89-90 - The participants were assessed while sitting and with the knee joint flexed at 90° and the ankle joint plantar dorsiflexed at 0°.
Suggest: The participants were assessed while sitting, with the knee joint flexed at 90° and the ankle joint at 0°. Also please add a reference.
→ Thank you very much for your indication. I deleted the word “planar dorsiflexed”. Regarding to a reference, previous studies reported the measurement of ultrasonography in prone position. In our study, I did not cite previous studies about the measurement position because lots of older people and patients could not lie in the prone position. So, I added the sentence as limitation into line 249-250 as follows:
First, ultrasonography was measured in the sitting position, which could be different from the measurement posture in other studies.
- Grip strength was measured using a digital dynamometer (T.K.K.5401; Takei Corp., Japan) in an upright position and defined as the highest value in the right or left hand for each trial.
How many was the test performed for each individual? Which one have you considered? Was any adaptation period? Meaning pre-repetitions aiming to get the subject familiar with the movement?
→ Thank you very much for your comments. Right and left grip strengths were measured once after first pretest. I added the sentence into line 107 as follows:
“and grip strength and walking speed were measured once after pretest.
- A stepwise multiple regression analysis was performed to construct the prediction model for SMI. The analysis incorporated SMI as the dependent variable and age, gender, BMI, grip strength, gait speed, and subcutaneous fat and gastrocnemius thicknesses as independent variables. Moreover, the residual analysis of the prediction model for SMI was analyzed using the Durbin–Watson test.
Why did you use Durbin-Watson? Please justify
→ Thank you very much for your question. The Durbin-Watson test can evaluate the validity of the prediction model made by multiple regression analysis, and the Durbin-Watson value is near 2 means that the residual error is small and the validity is high, so I used the Durbin-Watson test.
- Low gait speed and low grip strength is not clear. Please explain and describe it better in material and methods, for example how many tests have you performed?
→ Thank you very much for your comments. The measurements of gait speed and grip strength were defined as the highest value in the right or left hand for each trial, and grip strength and walking speed were measured once after pretest. I added the sentences into line 107 as follows:
Grip strength and walking speed were measured following the AWGS diagnostic criteria [4], and grip strength and walking speed were measured once after pretest.
The definitions of Low gait speed and low grip strength were written in line 111-113 in materials and methods. However, our explanation is hard to understand, so I added the definitions of low gait speed and low grip strength in table 2 of line 146-147.
- Adjusted R2 is not described neither in the legend nor in material and methods
→ Thank you very much for your indication. I have shown adjusted R2 of 0.48 in model 1, 0.77 in model 2, and 0.80 in model 3 in line 9 of table 3.
- Why did you not consider grip as a factor in the model? It had a positive relationship with SMI (table 2).
→Thank you very much for your question. In this study, I used the stepwise multiple regression analysis which includes factors of SMI automatically, and I wrote the sentences about why grip strength was excluded as a factor of SMI into line 217-221 as follows:
The previous study reported that physical performances were not necessarily related to SMI [9], physical performances could have been excluded as factors of SMI. In this study, gender was included as a factor of SMI, and the result of excluding grip strength as the factor could be affected by gender as a confounding factor because grip strength between gender had difference in table 1.
11.It would be interesting to have the individual values of SMI in parallel with the individual prediction values as a supplementary figure.
→ Thank you very much for constructive comments. I made the figure about actual and predicted SMI plots calculated by the stepwise multiple regression analysis as figure 3 in page 6 and added the sentence into line 166-177,” Figure 3 shows the plots of actual SMI and unstandardized predicted SMI”.
12.Mentioned as just on two lines in the results section and the very same statement is mentioned in the discussion section (was 2.03). What does the Durbin-Watson test mean? What message can the reader take from this test? How does it compare with the other analyses performed?
→ I’m sorry to confusing you. Because the sentences twice in results and discussion sections, I deleted the sentence of line 188-189” because the contribution rate of‥‥s 0.8, and the value of the Durbin–Watson test“.
I added and revised the sentence in line 164-167 about the explanation of the Durbin-Watson test as follows:
The statistic value of the Durbin–Watson test was 1.65 which was near 2, and only one participant was not included within ± 3SD (Figure 2). In addition, Figure 3 shows the plots of the actual SMI and unstandardized predicted SMI, and the validity of the prediction model was high.
Reviewer 2 Report
I appreciate the editors and the author the opportunity of reviewing this manuscript. In this regard, I think that the manuscript needs some work on the presentation - see comments attached.
The aim of the proposed manuscript was to construct a prediction model for skeletal muscle mass index in older adults Japanese using a non-invasive USG method for gastrocnemius thickness measurement.
Abstract
Line 10: Before using an abbreviation (SMI) please use the full name of the parameter
Introduction
I would suggest supplementing the introduction with a piece of short information on how sarcopenia influences life and why it is important to diagnose sarcopenia in the elderly.
Methods
Impedance is a method with error. A glass of water drunk before the test may affect the results. Were there any restrictions on fluid consumption prior to the study? What external conditions were at the time of the measurement (temperature, air humidity, were they monitored in some way).
General
Only two participants with sarcopenia engaged in the experiment lowers the value of work. The enrichment of the research with a larger group of participants with sarcopenia would indicate the usefulness of the proposed equation in the assessment of sarcopenia in the elderly. It would be an element of novelty. Diagnosing sarcopenia at an early stage using a simple, non-invasive method and implementing procedures to counteract its worsening (be it exercise or diet) could affect the quality of life of the elderly. The confirmation of the method, even on a small sample, would increase the value of the presented work
Author Response
Dear reviewer 2
Thank you very much for a lot of constructive advice for our manuscript in revision process.
I have answered for reviewer comments, added some sentences and revised our manuscript according to reviewer comments, and I have underlined at each sentence which I revised.
Line 10: Before using an abbreviation (SMI) please use the full name of the parameter
→ Thank you very much for your indication. I added the full name of SMI in line 10.
2.I would suggest supplementing the introduction with a piece of short information on how sarcopenia influences life and why it is important to diagnose sarcopenia in the elderly.
→ Thank you very much for your suggestion. I added the sentence into line 28,” leads to the decline of physical performances and malnutrition”, and into line 34,” to prevent progressing sarcopenia”.
3.Impedance is a method with error. A glass of water drunk before the test may affect the results. Were there any restrictions on fluid consumption prior to the study? What external conditions were at the time of the measurement (temperature, air humidity, were they monitored in some way).
→ Thank you very much for your comments. All participants were measured from 8:00 to 11:00 AM at the same place, and instructed not to eat breakfast. However, we could not monitor temperature, air humidity, and so on. I added the sentence about the condition of measuring SMI into line80-81 as follows:
All participants were instructed not eat breakfast, and measurements were conducted from 8:00 to 11:00 AM at the same place.
I also added the sentence into line 250-252 as study limitation as follows:
Second, during the measurement of SMI by bioelectrical impedance analysis, factors such as temperature and humidity could not be monitored.
3.Only two participants with sarcopenia engaged in the experiment lowers the value of work. The enrichment of the research with a larger group of participants with sarcopenia would indicate the usefulness of the proposed equation in the assessment of sarcopenia in the elderly. It would be an element of novelty. Diagnosing sarcopenia at an early stage using a simple, non-invasive method and implementing procedures to counteract its worsening (be it exercise or diet) could affect the quality of life of the elderly. The confirmation of the method, even on a small sample, would increase the value of the presented work
→ Thank you very much for constructive comments. I follow your advice, and exclude two sarcopenia participants and re-analyzed data in 193 participants.
According to the re-analysis, I revised the sentences and data as follows:
・Line 13: 195 →193
・Line 20: 0.1 →0.09
・Line 21: R=0.9 → 0.89
・Line 72: I added the sentence “ with sarcopenia”
・Line 74: 195 → 193.
・Line 140: women: n= 123→121.
・Table 1: Underlined data.
・Line 150: ρ=0.68 → 0.67
・Line 151: r=0.52 →0.51
・Table 2: Underlined data
・Line 161: 0.10 → 0.09
・Line 165: The Durbin-Watson test was 2.03 → 1.65
・Table 3:Underlined data
・In line 191-192, I deleted the sentences “Moreover, the number of participants with sarcopenia was 2 (1%)”.
・In line 254-256, I revised the sentences as follows:
Fourth, since the participants had no sarcopenia with relatively good physical performance, the adaptability of the prediction model for participants with sarcopenia is not proven.